# Investigating Genetic and Environmental Substrates of the Relationship between Positive Mental Health and Biological Aging—A Study Protocol

**DOI:** 10.3390/brainsci13121720

**Published:** 2023-12-15

**Authors:** Francesca Marcon, Miriam Salemi, Cristina D’Ippolito, Angelo Picardi, Virgilia Toccaceli, Lorenza Nisticò, Sabrina Alviti, Ester Siniscalchi, Francesca Salani, Giorgia Maria Varalda, Emanuela Medda, Corrado Fagnani

**Affiliations:** 1Unit of Mechanisms/Biomarkers/Models, Department of Environment and Health, Istituto Superiore di Sanità, 00161 Rome, Italy; francesca.marcon@iss.it (F.M.); ester.siniscalchi@iss.it (E.S.); francesca.salani@iss.it (F.S.); giorgia.varalda@iss.it (G.M.V.); 2Center for Behavioral Sciences and Mental Health, Istituto Superiore di Sanità, 00161 Rome, Italy; miriam.salemi@iss.it (M.S.); cristina.dippolito@iss.it (C.D.); angelo.picardi@iss.it (A.P.); virgilia.toccaceli@iss.it (V.T.); lorenza.nistico@iss.it (L.N.); sabrina.alviti@iss.it (S.A.); emanuela.medda@iss.it (E.M.)

**Keywords:** aging, mental health, telomere length, mitochondrial DNA, twin study, genetics, environment

## Abstract

Background: The Italian National Institute of Health (Istituto Superiore di Sanità) funded a 30-month project (July 2021–January 2024) to conduct a twin study of the relationships between Positive Mental Health (PMH) and cellular longevity. Only a few previous studies have focused on the biomarkers of aging in relation to psychological well-being, and none of them exploited the potential of the twin design. Method: In this project, following the standard procedures of the Italian Twin Registry (ITR), we aim to recruit 200 adult twin pairs enrolled in the ITR. They are requested to complete a self-report questionnaire battery on PMH and to undergo a blood withdrawal for the assessment of aging biomarkers, i.e., telomere length and mitochondrial DNA functionality. The association between psychological and aging biomarker measures will be assessed, controlling for genetic and familial confounding effects using the twin study design. Results and conclusions: Biomarker assays are underway. Once data are available for the total study sample, statistical analyses will be performed. The project’s results may shed light on new mechanisms underlying the mind–body connection and may prove helpful to promote psychological well-being in conjunction with biological functioning.

## 1. Introduction

### 1.1. Rationale and Novelty of Project Topic, Aims, Design, and Tools

Positive mental health (PMH) recently became a major target of scientific investigation, particularly in the biomedical field, because gaining insights into the complexity of factors influencing PMH could have implications for mental and physical comorbidity and, potentially, for the prevention and treatment of depression. Defining PMH is not an easy task. An authoritative review [1] listed a number of models that are relevant to the construct: above-normal functioning, the presence of multiple human strengths, maturity, the dominance of positive emotions, high social and emotional intelligence, subjective well-being, and resilience. Previous studies explored the biological correlates of psychological status variously measured [2]; most of these studies focused on markers of immunological activity, while only a minority of them targeted biomarkers of aging. It has been observed, for example, that posttraumatic stress disorder (PTSD) symptoms are associated with accelerated aging and that positive mental health moderates this relationship [3]. On the other hand, it has been shown that healthy mental aging may be influenced by the ownership of a pet, listening to music, and practice of meditation [4] and that volunteering is associated with a better self-perception of aging and positive mental health in later life [5].

Research on the biological mechanisms underlying the relationship between PMH and aging is still at an early stage [6,7]. One mechanism potentially involved could be biological aging. Biological age correlates with the chronological age of a person, but it is considered more reliable and reproducible than chronological age to investigate the effects associated with aging [8,9]. A key biological process in aging is cellular senescence, which is associated with stressors such as telomere shortening or mitochondrial dysfunctions. Thus, these biomarkers can be used as biological age predictors able to detect excessive acceleration or deceleration of aging and to highlight differences in the risk of age-related diseases for individuals of the same chronological age [8,10]. The use of telomere length as a biomarker of aging is based on the scientific knowledge of the processes of replication and repair of telomeres, which are defined as the biological clock of a cell. The progressive shortening of telomeres limits the number of replications of a cell and—once a critical value is reached—induces cellular senescence, a process associated with the aging of cells, tissues, and, therefore, organisms. The speed of telomere shortening is affected by the level of oxidative stress, which can be modulated by environmental and genetic factors as well as by the psycho-physical conditions of an organism [11]. The level of oxidative stress also influences mitochondrial function, and damaged mitochondria may, in turn, increase the production of oxygen free radicals and contribute to fuel oxidative stress. A close relationship between mitochondrial function and telomere length maintenance has been consistently demonstrated [12], also in population studies [13].

Interestingly, it was reported that the analysis of variations in telomere length and mitochondrial function can provide a reliable indication of the systemic effects associated with mental well-being [14,15]. Meta-analyses also demonstrated the association between primarily depression and telomere shortening, while the nature of this relationship has not been fully clarified [16].

Available studies in this line of research are far from providing a comprehensive picture of the phenomenon, as these studies concern only a few specific aspects of PMH, whereas some of the relevant PMH components outlined above were totally overlooked. Furthermore, published results may have been affected by serious methodological weaknesses, such as small sample sizes, doubtful generalizability, as well as low or absent control for genetic and environmental confounding factors that may have contributed to the observed association between PMH and biological aging and that we will be able to optimally control for in our study.

The objectives of this project are as follows: (i) to estimate the contribution of genetic, early-family, and individual-specific environmental factors to inter-individual differences in several PMH traits and in specific markers of biological aging; (ii) to explore the association between the PMH traits and the aging biomarkers, controlling for both known and unknown (i.e., genetic and familial) confounding factors; and (iii) to determine to what extent the observed association between the PMH traits and the aging biomarkers is explained by genetic and environmental factors that simultaneously affect PMH and biological aging.

For these purposes, a general population sample of adult twins enrolled in the Italian Twin Registry (ITR) [17] is recruited, and the genetically informative twin study design is applied. This design represents a major novelty in the research area concerning the biological correlates of PMH, as it was never used in previous investigations to address these issues. Based on the comparison of monozygotic (MZ) twins, genetically identical, with dizygotic (DZ) twins, who share 50% of their genetic background, the design allows for the estimation of genetic and environmental contributions to the expression of a given phenotype (or disease) and the co-expression (or co-morbidity) of multiple phenotypes (or diseases) [18].

### 1.2. Rationale of Targeted Positive Mental Health (PMH) Traits

According to Vaillant [1], the following constructs are considered relevant in PMH assessment: (i) positive psychological functioning; (ii) subjective well-being; (iii) balance between positive and negative emotions; (iv) emotion recognition and regulation; (v) attachment security; and (vi) ability to cope with stress. Within the project, this multi-dimensional PMH model is linked to existing knowledge about temperament and personality while separating the effects of positive and negative affect and exploring the possible mediating role of physical activity.

### 1.3. Rationale of Targeted Aging Biomarkers

This project plans to correlate positive mental status with biological aging via the analysis of telomere length and mitochondrial function, biomarkers significantly correlated to the biological age of individuals [8,10]. The use of these two correlated biomarkers will reinforce the biological plausibility of the outcomes.

Blood has been selected as the biological material for the analyses of aging biomarkers. This choice was mainly dictated by the following considerations:

(i) The study is aimed at evaluating if psychological well-being is associated with healthy aging, and this latter component is assessed not only by the speed of aging, obtained from the comparison between biological and chronological age but also by blood chemistry analyses.

(ii) Although it is generally recognized that the analysis of biomarkers of aging in a specific tissue and the extrapolation of the general age of the organism from this analysis can represent a limitation, it has also been observed that blood cells represent a reliable surrogate tissue for studying systemic effects associated with aging and psychological well-being [14].

(iii) Blood is an easily accessible biological material whose withdrawal usually does not cause stress in the enrolled subjects.

Overall, the study intends to explore systemic effects associated with mental well-being, and the analysis of variation in telomere length and mitochondrial functionality represent reliable biomarkers of these effects [11].

## 2. Materials and Methods

The activities of the present two-year project are scheduled as follows:

Month 1–6: Recruitment of study subjects based on the standard procedures of the Italian Twin Registry (ITR). Collection of self-reported positive mental health (PMH) data.

Month 7–12: Collection of biological samples. Biological assays. DNA-based assessment of zygosity of same-gender pairs. Face-to-face re-collection of PMH data on a subsample of the total sample of twin pairs, according to funds and time available.

Month 13–18: Statistical analysis of PMH and biological markers data.

Month 19–24: Drafting of a first paper and dissemination of results via the ITR website and social media platforms.

### 2.1. PMH Test Battery

The following self-report psychological test battery is administered to participants: (i) Ryff scales [19] and Empathy Quotient [20] (for positive psychological functioning); (ii) Rosenberg scale [21], Satisfaction with Life scale [22], and Life Orientation Test [23] (for subjective well-being); (iii) Positive and Negative Affect Schedule [24] (for balance between positive and negative emotions); (iv) Toronto Alexithymia Scale [25] (for emotion recognition and regulation); (v) Experiences in Close Relationships questionnaire [26] (for attachment security); (vi) Dispositional Resilience Scale [27] (for the ability to cope with stress); (vii) Cohen’s Perceived Stress Scale [28] (for perceived stress); (viii) Temperament and Character Inventory [29] and Big Five Inventory [30] (for temperament and personality); and furthermore, (ix) the International Physical Activity Questionnaire [31] is used to explore the mediating role of physical activity.

### 2.2. Collection of Biological Samples

Biological samples are collected at a reference laboratory, transferred to the Italian National Institute of Health (Istituto Superiore di Sanità, ISS) within two hours from collection, processed as described below, and appropriately stored in the Italian Twin Registry (ITR) biobank.

#### 2.2.1. Whole Blood

Whole blood is collected by venipuncture withdrawal using different types of vacutainers based on the specific subsequent applications.

#### 2.2.2. DNA Extraction

Blood is aliquoted (600 μL) in 1.5 mL vials (Eppendorf safe-lock tubes) after collection in an EDTA vacutainer (Becton Dickinson, Milano, Italy) and stored at −20 °C. DNA is purified using the Blood Isolation mini kit (Norgen, Biotech Corp., Thorold, ON, Canada) following the manufacturer’s instructions. The quality and quantity of DNA are measured using a Nanodrop spectrophotometer (Euroclone, Milano, Italy). DNA is used to perform the analysis of telomere length and mitochondrial DNA copy number (mtDNAcn).

#### 2.2.3. RNA Extraction

Blood is aliquoted (500 μL) in 2 mL vials (Eppendorf safe-lock tubes) after collection from EDTA vacutainer (Becton Dickinson, Milano, Italy), immediately diluted 1:1 in RNA later solution (Invitrogen, Thermo Fisher Scientific, Milano, Italy) and stored at −20 °C in the biobank. RNA is purified using the Blood Isolation mini kit (Norgen, Biotech Corp., Thorold, ON, Canada) following the manufacturer’s instructions. The quality and quantity of RNA are measured using a Nanodrop spectrophotometer.

#### 2.2.4. Serum Collection

Whole blood, collected in clot-activator tubes (Becton Dickinson Vacutainer, Milano, Italy), is processed within two hours from withdrawal. Samples are centrifuged at 2500 rpm for 10 min, room temperature (RT). Following centrifugation, the serum is immediately aliquoted (500 μL) into 1.5 mL clean polypropylene tubes (Eppendorf safe-lock tubes, Milano, Italy) and stored at −80 °C in the biobank.

#### 2.2.5. Plasma Collection

Whole blood, collected into EDTA-treated tubes (Becton Dickinson Vacutainer), is centrifuged at 2500 rpm for 10 min, RT. The resulting plasma is aliquoted (500 μL) into 1.5 mL clean polypropylene tubes (Eppendorf safe-lock tubes) and stored at –80 °C in the biobank.

#### 2.2.6. PBMC Isolation

Whole blood (8 mL) is collected in sodium heparin tubes (Becton Dickinson Vacutainer). Peripheral blood mononuclear cells (PBMCs) are isolated using Histopaque-1077 gradient (Sigma-Aldrich, Milano, IT). Briefly, blood diluted 1:1 in isotonic phosphate-buffered saline solution is layered onto Hystopaque-1077 in a ratio of 2:1 and immediately centrifuged at 1400 rpm for 30 min, RT, without brake. PBMCs at the plasma/Histopaque-1077 interface are collected and transferred into a clean conical tube, washed two times in 10 mL of PBS by centrifugation at 1000 rpm for 10 min at RT. To determine the number and vitality of the cells, 10 μL of cell suspension (PBMCs) is diluted in 1:1 with Trypan blue exclusion dye solution (Sigma-Aldrich, cod. T8154, Milano, Italy) and counted using cell counting chamber slides (Kova Plastics Glasstic Slides, 10 grids, Kova International, Garden Grove, CA, USA). Cells are frozen in 90% inactivated fetal bovine serum (Sigma-Aldrich, Milano, Italy) with 10% dimethyl sulfoxide (DMSO) at a density of 10 × 10^6^/mL. Cryotubes (Biologix Group Limited, cod. 81-8204, Hallbergmoos, Germany) are stored in liquid nitrogen in the biobank.

### 2.3. Sample Size and Recruitment

Target sample size: 200 twin pairs.

Inclusion criteria: twins previously enrolled in the ITR [17], residents in Rome and its province, aged 18–80 years, males and females, monozygotic (MZ) and dizygotic (DZ), and with available and updated email addresses.

Exclusion criteria: previous diagnosis of major medical or psychiatric diseases.

Regarding the targeted age range, it is important to highlight that the main study objective is to verify if and to what extent mental status can affect cellular aging (or cellular longevity, as the other side of the coin), with the latter processes acting in a “continuous” way. The effects of these processes can be observed all life long, and for this reason the present project focuses on these effects across the entire adulthood span. Using this approach, for example, it will be possible to capture differences in aging profiles, even between young adult individuals, if these individuals present with varying psychological profiles.

This study has adequate statistical power to test the hypotheses of interest. For moderately heritable (40%) traits, such as various psychological and biological phenotypes, a sample size of 200 twin pairs provides 80% power (alpha 0.05) to reject the null hypothesis of the absence of genetic effects. Furthermore, our planned sample size confers more than 80% power (alpha 0.05) to estimate correlations of small to moderate magnitude (r > 0.28) between psychological and biological measures, which is the expected magnitude for correlations among the study parameters. Power calculations for genetic estimates were performed with the software Mx [32], as described by Neale and Cardon [33]. More in detail, these calculations were carried out by fitting the known model to the exact (population) covariance matrices. Constraining a certain set of parameters to zero and refitting the model provides the non-centrality parameter related to that particular constraint. From this non-centrality parameter, the sample size required to reject the constrained (i.e., false) model with a probability (i.e., a power) of 0.80 and a significance level of 0.05 can be calculated and is supplied by Mx. The target sample size of 200 twin pairs is in line with almost all twin studies previously conducted by our group at the ITR, which focused on a large variety of complex phenotypes, including psychological and personality traits.

The twin pairs are recruited following the standard procedures of the ITR [17]. Briefly, eligible twins are contacted by email and are asked about their potential willingness to be involved in the study and to receive further information by telephone. They are, in fact, requested to provide a phone number in the email response. Agreeing twins are mailed an informed consent form regarding study participation, use of participant’s data and biological sample for this specific study, and biological sample storage in the ITR biobank for future studies; furthermore, they are mailed an informative letter explaining the study aims and procedures, questionnaires on socio-demographic, lifestyle, and health information (including major clinical and psychiatric conditions), as well as the PMH test battery (see above). After acceptance of participation, these twins are contacted by telephone to fix an appointment at the reference laboratory (located in Rome) for a blood withdrawal; the appointment is fixed at least two weeks after the shipment of the study material. On the occasion of the blood withdrawal, participants bring the signed informed consent and the filled questionnaires back to the ITR staff, and possible criticisms (e.g., inconsistent or missing information) are resolved in a face-to-face setting. The blood samples are collected by trained personnel according to a standard protocol internal to the laboratory. These samples are used to assess aging biomarkers (see below) and to perform routine check-up analyses (i.e., full blood count, glucose, creatinine, total proteins, GOT, GPT, cholesterol, triglycerides). The results of check-up analyses, which are released for free to participants, allow the ITR staff to obtain general health information to be used in the assessment of the association between PMH and aging biomarkers.

The procedures implemented in this specific study, as all research activities performed by the ITR, conform to Italian laws and to the European General Data Protection Regulation (GDPR) with respect to the collection, use, sharing, and storage of personal information. Briefly, personal data are treated electronically by authorized personnel in compliance with confidentiality and security criteria and are used exclusively for the objectives of this study. Socio-demographic and health information are stored in pseudonymized (i.e., unrecognizable) form, separately from identification data. Collected data and individual results will not be communicated to third parties outside the project and will be published only in aggregate form for scientific research finalities.

### 2.4. Biomarker Assays

#### 2.4.1. Analysis of Mitochondrial DNA Copy Number

Samples of DNA are diluted to obtain the working concentration of 4 ng/µL to be used for each experimental point carried out in triplicate. The amount of mtDNA is measured using quantitative polymerase chain reaction (qRT-PCR) to determine the levels of the nuclear gene HBG (human beta-globin) and the mitochondrial gene ND1. The amplification of HBG is obtained using the following primers: (forward) 5′-GAAGAGCCAAGGACAGGTAC-3′, (reverse) 5′-CAATTCATCCACGTTCACC-3′, while for ND1, the primers used are as follows: (forward) 5′-AACATACCCATGGCCAACCT-3′, (reverse) 5′-AGCG-AAGGGTTGTAGTAGCCC-3′. The amplification reaction contains 0.6 µL of each primer (working solution 10 µM; final concentration 300 nM), 10 µL of SYBR Green PCR Master Mix (Bioline Meridian Bioscience, London, UK), 5 µL of genomic DNA (working solution 4 ng/µL; final concentration 20 ng/well) and purified water to a total volume of 20 µL. A standard curve, a negative control (no DNA template), and a reference sample of genomic DNA are included in each experiment. For the standard curve, a sample of genomic DNA is diluted serially to produce five final concentrations (20, 10, 5, 2.5, 1.25 ng/well). The PCR cycling conditions for both amplicons are 95 °C for 10 min, followed by 40 cycles at 95 °C for 15 s and 60 °C for 1 min. The specificity of the PCR reaction is checked via the analysis of melting curves obtained at the end of each PCR. RT-PCR is performed using the ABI Prism 7500 Sequence Detection System (Applied Biosystems, Monza, MB, IT). Fluorescence is analyzed using the ABI Prism 7500 FAST SDS software version 2.0 year 2010 to quantify PCR products for each sample based on the standard curve. The amount of mtDNA is obtained by the ratio between the copy number of the mitochondrial gene ND1 and the nuclear gene HBG (human beta-globin). The ratio of mtND1/HBG is normalized using the reference sample of genomic DNA. The same position for each sample (i.e., standard curve, reference sample, test samples) is maintained on the two plates (nuclear HBB and mitochondrial ND1 reactions).

#### 2.4.2. Analysis of Telomere Length

Telomere length is measured by quantitative real-time PCR (qRT-PCR) using DNA samples purified from whole blood as described above. This method is based on the rationale that the amount of telomere signal per genome measured by qPCR represents the average telomere length in a given DNA sample [34]. Telomere length is quantified as the relative ratio of telomere (T) repeat copy number to a single copy gene (S), called the T/S ratio, in experimental samples using standard curves. The 36B4, encoding acidic ribosomal phosphoprotein P0, is used as the control single copy gene needed to quantify input genomic DNA and to normalize the signal from the telomere reaction. The primer sequences for telomere amplification are (forward TelF) 5′-GGTTTTTGAGGGTGAGGGTGAGGGTGAGGGTGAGGGT-3′; (reverse TelR) 5′-TCCCGACTATCCCTATCCCTATCCCTATCCCTATCCCTA-3′. The primer sequences for the amplification of the reference gene 36B4 are (forward 36B4F) 5′CAGCAAGTGGGAAGGTGTAATCC3′; (reverse 36B4R) 5′CCCATTCTATCATCAACGGGTACAA3′. The reaction is carried out in triplicate. The PCR master mix includes 10 µL of SensiFAST SYBR Hi-ROX master mix (Bioline Meridian Bioscience, London, UK), 2 µL of forward primer and 2 µL of reverse primer (working solution 1 µM; final concentration 100 nM), 5 µL of genomic DNA (working solution 4 ng/µL; final concentration 20 ng/well) and purified water to a total volume of 20 µL. A standard curve and a negative control (no DNA template) are included in each experiment. For the standard curve, a reference DNA sample is diluted serially to produce six final concentrations (20, 10, 5, 2.5, 1.25, 0.625 ng/well). The PCR cycling conditions for both amplicons are 95 °C for 20 s, followed by 40 cycles at 95 °C for 3 s and 60 °C for 30 s. The specificity of the PCR reaction is checked via the analysis of melting curves obtained at the end of each PCR. RT-PCRs are performed using the ABI Prism 7500 Sequence Detection System (Applied Biosystems, Monza, MB, IT). Fluorescence is analyzed with the ABI Prism 7500 FAST SDS software version 2.0 year 2010 to quantify PCR products for each sample based on the standard curve. The resulting T/S ratio represents the average telomere length per genome. The same position for each sample (i.e., standard curve, test samples) is maintained on the two plates (telomere and 36B4 reactions).

### 2.5. Statistical Analysis of PMH and Biomarker Data

The statistical analysis plan includes (i) Psychometric analyses (i.e., Confirmatory Factor Analyses, Item Response Theory modeling, reliability and validity analyses). (ii) Latent variable models to estimate PMH dimensions scores and to model PMH as a higher-order factor. (iii) Quantitative genetic models [33] to decompose the variance of the parameters and the covariance among them into genetic and environmental components; these models allow for estimating the heritability of a trait and the genetic correlation between traits (Figure 1) and examining gene-environment or gene-age interactions by testing for differences in the genetic estimates across strata of a putative exposure factor or across age-groups. (iv) Cotwin-control analysis, looking for an association between the within-pair difference in PMH traits scores and the within-pair difference in biomarkers measures; this analysis is extremely powerful as it exploits a built-in matching by unknown genetic, intrauterine, and early-familial factors, as well as by known confounders like age. The analyses (iii) and (iv) are unique to the twin study setting and represent valuable advantages over the classical matched designs. Although twin pair matching increases control over genetic and shared environmental factors, it is unable to control for unshared confounders that only affect one twin in the pair; these individual-specific factors still need to be included as covariates in the models of analysis.

### 2.6. Results and Interpretation

Besides estimating genetic and environmental influences on psychological functioning and biological aging separately, which is already by itself an important achievement, this project is expected to generate novel evidence on the association between PMH and cellular longevity and on the factors underlying this association. The twin study design offers a unique opportunity to discern whether the targeted PMH constructs and aging biomarkers share—at least in part—the same genetic and environmental effects or they are possibly connected by direct causal paths. In this respect, if the association in the total sample happens to decline or even disappear when optimally controlling for genetic and environmental confounding factors within MZ twin pairs, then “third” intervening factors are likely to explain the association; on the contrary, if the association persists within MZ twin pairs, then a causal dependence should be seriously considered. Clarifying the nature of the relationship between psychological well-being and biological longevity is essential for any intervention aimed to foster both human health components at the same time.

### 2.7. Project Dissemination

To achieve adequate participant enrolment and to disseminate the project’s objectives, overall findings, and their potential relevance in terms of etiological and public health implications to the general public and to the scientific community, the standard strategies and tools of the ITR are followed. These include the institutional ITR website, in which a project-specific subsection has been implemented, as well as the ITR Facebook and Instagram pages; furthermore, the Press Office of the ISS periodically releases official communications. For this specific study, a task force was created to encourage participation and resolve any doubts or fears by telephone among eligible twins. The most relevant results generated by the project are planned to be presented at international congresses and workshops, and to be published in highly ranked scientific journals.

## 3. Summary and Conclusions

The project’s relevance stems from three main aspects. First, the results may have high theoretical value, adding missing pieces of knowledge to the complex puzzle of the interplay between the mind and body and drawing attention to novel mechanisms possibly linking mental and physical health. Second, by providing quantification of the environmental overlap between PMH and biological aging, as assessed by the targeted measures and markers, our study may encourage the search for exposure factors that may positively influence both the psychological status and biological dynamics of aging. The identification of these factors may allow the designing of intervention programs intended to promote psychological well-being and biological functioning at the same time, rather than approaching them separately. Third, by estimating if and to what extent PMH and biological aging share common genetic substrates, our effort may foster subsequent research aimed at better characterizing genetic risk for mental disease in conjunction with problematic biological processes related to aging.

Possible weaknesses of this project are related to the arbitrary definition of PMH, the questionable reliability of self-reported PMH data, and the doubtful short-term transferability of results. However, an unprecedented variety of constructs that are relevant to PMH is evaluated according to the guidelines provided by an authoritative review [1]. Furthermore, to increase the reliability of collected data for a subset of the proposed PMH scales, self-report information is planned to be complemented with face-to-face interviews administered to study participants by the ITR staff at the moment of blood sample collection. Finally, although the project’s results may not be easily seen as useful in the short run, they may provide further compelling evidence in favor of the so-called “positive psychology approach”, which is still used only sporadically in clinical practice.

## Figures and Tables

**Figure 1 brainsci-13-01720-f001:**
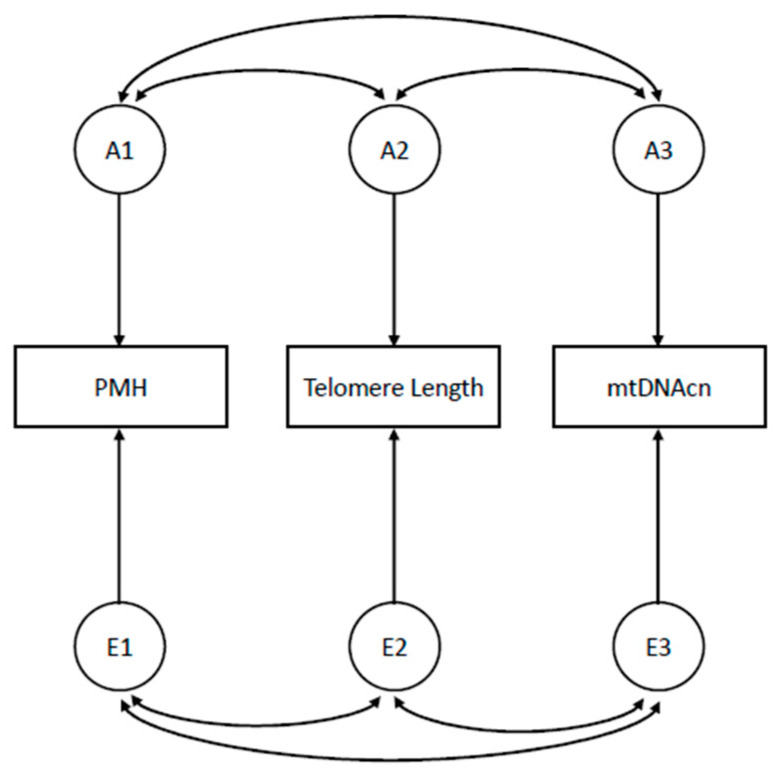
Correlated Factors Model for the estimation of genetic and environmental correlations between PMH traits and aging biomarkers. Note: Ai, genetic factors influencing study variables; Ei, environmental factors influencing study variables; PMH, Positive Mental Health trait; mtDNAcn, mitochondrial DNA copy number; double-headed arrows are correlations between (genetic or environmental) factors influencing study variables.

## Data Availability

The data are not publicly available due to privacy or ethical restrictions. The datasets generated during the current study will be made available from the corresponding author upon reasonable request.

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
