# Peer review of "Investigating Genetic and Environmental Substrates of the Relationship between Positive Mental Health and Biological Aging—A Study Protocol"

_brainsci, 2023, doi:10.3390/brainsci13121720_

Round 1

Reviewer 1 Report (Previous Reviewer 1)

Comments and Suggestions for Authors

A novel study. My main concerns are:

1) PMH appears to be an ill defined construct as highlighted. The assessment of PMH using 12 scales seems unrealistic for the participant. How long is this survey as in time and number of questions? How does factor analysis help in PMH assessment? Is it that some scales or items within each scale may be suppressed for analysis? If so would the scales used lose validity? 

2) How do you control for the myriad of factors that can explain PMH? Is 200 pairs enough considering that you may have to adjust for so many other confounders? 

3) What would these findings translate to in the real world if an association between PMH and telomere length/mtDNA is found? Is telomere length/mtDNA amenable to interventions. If these variables cannot be modified then interventions to improve PMH would be independent of telomere length/mtDNA.

Author Response

Responses to Reviewer 1

A novel study. My main concerns are:

1) PMH appears to be an ill defined construct as highlighted. The assessment of PMH using 12 scales seems unrealistic for the participant. How long is this survey as in time and number of questions? How does factor analysis help in PMH assessment? Is it that some scales or items within each scale may be suppressed for analysis? If so would the scales used lose validity?

Response

In the present project, we defined the multi-faceted construct of PMH according to an authoritative review (Vaillant 2012, referenced in the manuscript) that listed a number of models relevant to the construct. This definition encompasses the main components of wellbeing, and it is not necessarily linked to disease.

The scales we propose in our project will allow us assessing PMH with an unprecedented level of diversity and comprehensiveness. We experienced in previous surveys promoted by the Italian Twin Registry that such a wide test battery can be administered to samples of hundreds of twin pairs within a period of few months, and this is exactly what this project aims to do. Indeed, as we have implemented this protocol in January 2021 (soon after the Ethics Board approval) and we have decided only recently to share the protocol with the scientific community, subjects’ recruitment (i.e., 200 twin pairs) and data collection have been completed during the preparation of this paper, and statistical analyses of both psychological and biological data are underway.

As regards this latter stage, the various scales are being analysed both separately, to identify the specific PMH aspects more strongly correlated with biological ageing, and globally, as proxies of a higher-order “PMH” factor. The advantages of considering higher-order latent factors are well-known and include the ability to isolate scale-specific measurement errors by estimating error variances. These techniques are routinely used in psychological and biomedical research, and we believe they do not need to be expanded further in a study protocol.                   

2) How do you control for the myriad of factors that can explain PMH? Is 200 pairs enough considering that you may have to adjust for so many other confounders?

Response

As described in the text, the twin study design offers a unique context in which several known and unknown confounders can be taken under control by a strong matching (within twin pairs) for intra-uterine as well as genetic and shared familial factors. This is just one of the main novelties of our study compared to previous observational ones, which are generally much more prone to residual confounding effects when applying classical methods of adjustment. However, it remains hard to control for unshared (individual-specific) factors, and this represents one limitation of the twin study approach. We have briefly mentioned this aspect in the text (“Statistical analysis of PMH and biomarker data”). AS regards sample size, power calculations were performed and reported in the text, showing that our sample size of 200 twin pairs is adequate to estimate the parameters of interest in this project. 

3) What would these findings translate to in the real world if an association between PMH and telomere length/mtDNA is found? Is telomere length/mtDNA amenable to interventions. If these variables cannot be modified then interventions to improve PMH would be independent of telomere length/mtDNA.

Response

Of course, biomarkers measures are not directly modifiable. However, as it is explained in the Conclusions and it is stressed in most cited literature, a possible association estimated between PMH and biomarkers of ageing would represent valuable evidence that interventions to promote PMH are likely to have beneficial effects also on ageing processes.

Reviewer 2 Report (Previous Reviewer 2)

Comments and Suggestions for Authors

I would like to thank the editors and authors for the opportunity to review the manuscript “nvestigating Genetic and Environmental Substrates of the Relationship between Positive Mental Health and Biological Aging – A Study Protocol”

The theme chosen is interesting and of greater importance, knowing that positive mental health and well-being play an important role in overall health. Research has objectively demonstrated a positive association between positive mental health dimensions such as resilience, optimism and social commitment, among others, with better health outcomes and increased longevity and well-being. However, there are not many studies focused on biomarkers of aging such as telomere dynamics and mitochondrial DNA copy number. In this sense, this study can contribute to encouraging the search for factors that can positively influence both the psychological state and the biological dynamics of aging.

I will now offer my contributions or suggestions for improving the text:

2. Materials and methods

It seems to me that the authors should clarify:

·       Participant inclusion and exclusion criteria and eligibility criteria.

·       Strategies were thought of to improve adherence to the research protocol by participants.

·       Include a study schedule, how long, or when you plan to develop, or a schematic diagram-

·       How the estimated number of participants needed to achieve the study objectives and any sample size calculations were determined.

·       Strategies for Achieving Adequate Participant Enrollment to Reach Target Sample Size

·       What are the plans to obtain ethics committee/institutional review board approval?

·       Confidentiality - Who and how will obtain informed consent to collect and use participant data and biological samples.

·       Clarify plans for data entry, coding, security, and storage. How personal information about potential and enrolled participants will be collected, shared, and maintained to protect confidentiality before, during, and after the study.

·       Financial and other competing interests for the principal investigators of the overall trial and each study site

·       Declaration of who will have access to the final trial dataset and disclosure of contractual agreements that limit this access for investigators.

Final decision:

The manuscript needs minor changes.

I hope that my contributions will serve to improve this article and the study you propose.

 Thank you very much.

Best Regards

Author Response

Responses to Reviewer 2

Materials and methods

It seems to me that the authors should clarify:

  • Participant inclusion and exclusion criteria and eligibility criteria.

Response: These aspects are already reported in the text but are now schematically arranged to make them more easily detectable by the reader.

  • Strategies were thought of to improve adherence to the research protocol by participants.

Response: The most important stages of the study protocol are self-report of psychological test battery and blood collection. As regards self-report of scales, in the occasion of blood withdrawal the questionnaires are reviewed by the research staff and possible criticisms are resolved in a face-to-face setting. With respect to blood withdrawal, it is performed by trained personnel according to a standard protocol, internal to the reference laboratory. These aspects are now better specified.  

  • Include a study schedule, how long, or when you plan to develop, or a schematic diagram-

Response: A study schedule has been included as requested.

  • How the estimated number of participants needed to achieve the study objectives and any sample size calculations were determined.

Response: Sample size calculations are already reported in detail as these calculations had been requested by a previous reviewer. In the current version of the protocol, we have stressed that the target sample size is in line with almost all twin studies previously conducted by our group.

  • Strategies for Achieving Adequate Participant Enrollment to Reach Target Sample Size

Response: Communications and dissemination strategies follow the standard procedures of the Italian Twin Register and are described in the “Project dissemination” section, which has been slightly amended to better comply with reviewer’s request.

  • What are the plans to obtain ethics committee/institutional review board approval?

Response: Although we have decided only recently to share this protocol with the scientific community, we have already obtained (on 21 January 2023) the approval of our funded project by the Ethics Board of the Italian National Institute of Health (Istituto Superiore di Sanità) and the study is now close to the end. The declaration about the Ethics Board approval is included at the bottom of the paper (before References).

  • Confidentiality - Who and how will obtain informed consent to collect and use participant data and biological samples.

Response: This aspect is already described in the “Sample size and recruitment” section with two sentences that we have now slightly improved.

  • Clarify plans for data entry, coding, security, and storage. How personal information about potential and enrolled participants will be collected, shared, and maintained to protect confidentiality before, during, and after the study.

Response: These aspects are part of the policies that regulate all research activities of the Italian Twin Registry, mostly within the European framework of the General Data Protection Regulation (GDPR). These aspects can hardly be described in this paper in an exhaustive manner, and for this reason we have included a paragraph at the end of the “Sample size and recruitment” section to briefly touch upon personal data treatment.   

  • Financial and other competing interests for the principal investigators of the overall trial and each study site

Response: Researchers involved in this study do not have any competing interests, as already declared at the end of the paper.

  • Declaration of who will have access to the final trial dataset and disclosure of contractual agreements that limit this access for investigators.

Response: As specified in the data treatment paragraph (see previous point), only authorized personnel can have access to the final dataset. These personnel are listed in the document that has been approved by the Ethics Board.

This manuscript is a resubmission of an earlier submission. The following is a list of the peer review reports and author responses from that submission.

Round 1

Reviewer 1 Report

Comments and Suggestions for Authors

A novel study. My main concerns are:

1) PMH appears to be an ill defined construct as highlighted. The assessment of PMH using 12 scales seems unrealistic for the participant. How long is this survey as in time and number of questions? How does factor analysis help in PMH assessment? Is it that some scales or items within each scale may be suppressed for analysis? If so would the scales used lose validity? 

2) How do you control for the myriad of factors that can explain PMH? Is 200 pairs enough considering that you may have to adjust for so many other confounders? 

3) What would these findings translate to in the real world if an association between PMH and telomere length/mtDNA is found? Is telomere length/mtDNA amenable to interventions. If these variables cannot be modified then interventions to improve PMH would be independent of telomere length/mtDNA.

Reviewer 2 Report

Comments and Suggestions for Authors

I would like to thank the editors and authors for the opportunity to review the manuscript “nvestigating Genetic and Environmental Substrates of the Relationship between Positive Mental Health and Biological Aging – A Study Protocol”

In general terms, I can say that the article presents specific references of interest, however, only seven out of 27 references are less than 5 years old.

The theme chosen is interesting and of greater importance, knowing that positive mental health and well-being play an important role in overall health. Research has objectively demonstrated a positive association between positive mental health dimensions such as resilience, optimism and social commitment, among others, with better health outcomes and increased longevity and well-being. However, there are not many studies focused on biomarkers of aging such as telomere dynamics and mitochondrial DNA copy number. In this sense, this study can contribute to encouraging the search for factors that can positively influence both the psychological state and the biological dynamics of aging.

 I will now offer my contributions or suggestions for improving the text:

1. Introduction

A greater description of the research question and its justification for carrying out the study is suggested, including summaries of relevant studies on the topic with more recent references (only seven out of 27 references are less than 5 years old). There is a model of positive mental health that I think the authors could explore and analyze to see if it makes sense for the study, as well as the positive mental health questionnaire by author Teresa Lluch, see some articles:

·       Lluch, M.T. (2003). Construcción y análisis psicométrico de un cuestionario para evaluar la salud mental positiva. Psicología Conductual. Revista internacional de psicología clínica y de la salud, 11(1) 61-78

·       Lluch‐Canut, M. T. (1999). Construccion de una escala para evaluar la salud mental positiva. (Doctoral Thesis, Department of Behavioral Sciences Methodology: University of Barcelona, Spain). Retrieved from http://dipos it.ub.edu/dspac e/bitst ream/2445/42359/ 1/E_TESIS.pdf.

·       Sequeira CAC, Pires MF, Carvalho JCM, Ribeiro IMOC, Lluch i Canut MT. (2021) Characteristics of a positive mental health program for adults: a Focus Group study. Revista Portuguesa de Enfermagem de Saúde Mental, 26. https://doi. org/10.19131/rpesm.319

·       Teixeira SMA, Coelho JCF, Sequeira CAC, Lluch i Canut MT, Ferré-Grau C. (2019). The effectiveness of positive mental health programs in adults: A systematic review. Health Soc Care Community, 00, 1–9. https://doi. org/10.1111/hsc.12776 Teixeira SMA, Ferré-Grau C,

2. Materials and methods

It seems to me that the authors should clarify:

·       Participant inclusion and exclusion criteria and eligibility criteria.

·       Strategies were thought of to improve adherence to the research protocol by participants.

·       Include a study schedule, how long, or when you plan to develop, or a schematic diagram-

·       How the estimated number of participants needed to achieve the study objectives and any sample size calculations were determined.

·       Strategies for Achieving Adequate Participant Enrollment to Reach Target Sample Size

·       What are the plans to obtain ethics committee/institutional review board approval?

·       Confidentiality - Who and how will obtain informed consent to collect and use participant data and biological samples.

·       Clarify plans for data entry, coding, security, and storage. How personal information about potential and enrolled participants will be collected, shared, and maintained to protect confidentiality before, during, and after the study.

·       Financial and other competing interests for the principal investigators of the overall trial and each study site

·       Declaration of who will have access to the final trial dataset and disclosure of contractual agreements that limit this access for investigators.

Final decision:

The manuscript needs minor changes.

I hope that my contributions will serve to improve this article and the study you propose.

 Thank you very much.

Best Regards